# Association of Human *FOS* Promoter Variants with the Occurrence of Knee-Osteoarthritis in a Case Control Association Study

**DOI:** 10.3390/ijms20061382

**Published:** 2019-03-19

**Authors:** René Huber, Holger Kirsten, Annu Näkki, Dirk Pohlers, Hansjörg Thude, Thorsten Eidner, Matthias Heinig, Korbinian Brand, Peter Ahnert, Raimund W. Kinne

**Affiliations:** 1Experimental Rheumatology Unit, Department of Orthopedics, Jena University Hospital, Waldkrankenhaus ‘Rudolf Elle’, 07607 Eisenberg, Germany; huber.rene@mh-hannover.de (R.H.); d.pohlers@skc.de (D.P.); raimund.w.kinne@med.uni-jena.de (R.W.K.); 2Institute of Clinical Chemistry, Hannover Medical School, 30625 Hannover, Germany; brand.korbinian@mh-hannover.de; 3Institute of Medical Informatics, Statistics, and Epidemiology, Universität Leipzig, 04107 Leipzig, Germany; pahnert@imise.uni-leipzig.de; 4LIFE–Leipzig Research Center for Civilization Diseases, Universität Leipzig, 04103 Leipzig, Germany; 5Translational Centre for Regenerative Medicine (TRM), Universität Leipzig, 04103 Leipzig, Germany; 6Fraunhofer Institute for Cell Therapy and Immunology IZI, 04103 Leipzig, Germany; 7Institute for Molecular Medicine Finland FIMM, University of Helsinki, 00014 Helsinki, Finland; annu.posti@gmail.com; 8Public Health Genomics Unit, National Institute for Health and Welfare, 00271 Helsinki, Finland; 9Institute of Transfusion Medicine, Jena University Hospital, 07747 Jena, Germany; h.thude@uke.de; 10Department of Internal Medicine III, Division of Rheumatology & Osteology, Jena University Hospital, 07747 Jena, Germany; thorsten.eidner@med.uni-jena.de; 11Department of Computational Molecular Biology, Max Planck Institute for Molecular Genetics, 14195 Berlin, Germany; matthias.heinig@helmholtz-muenchen.de

**Keywords:** AP-1, JUN, FOS, promoter, knee osteoarthritis

## Abstract

Our aim was to analyse (i) the presence of single nucleotide polymorphisms (SNPs) in the *JUN* and *FOS* core promoters in patients with rheumatoid arthritis (RA), knee-osteoarthritis (OA), and normal controls (NC); (ii) their functional influence on *JUN*/*FOS* transcription levels; and (iii) their associations with the occurrence of RA or knee-OA. *JUN* and *FOS* promoter SNPs were identified in an initial screening population using the Non-Isotopic RNase Cleavage Assay (NIRCA); their functional influence was analysed using reporter gene assays. Genotyping was done in RA (*n* = 298), knee-OA (*n* = 277), and NC (*n* = 484) samples. For replication, significant associations were validated in a Finnish cohort (OA: *n* = 72, NC: *n* = 548). Initially, two SNPs were detected in the *JUN* promoter and two additional SNPs in the *FOS* promoter in perfect linkage disequilibrium (LD). *JUN* promoter SNP rs4647009 caused significant downregulation of reporter gene expression, whereas reporter gene expression was significantly upregulated in the presence of the *FOS* promoter SNPs. The homozygous genotype of *FOS* promoter SNPs showed an association with the susceptibility for knee-OA (odds ratio (OR) 2.12, 95% confidence interval (CI) 1.2–3.7, *p* = 0.0086). This association was successfully replicated in the Finnish Health 2000 study cohort (allelic OR 1.72, 95% CI 1.2–2.5, *p* = 0.006). *FOS* Promoter variants may represent relevant susceptibility markers for knee-OA.

## 1. Introduction

Rheumatic diseases, e.g., rheumatoid arthritis (RA) and knee-osteoarthritis (OA), are characterized by inflammation and destruction of multiple joints. In RA, activated synovial fibroblasts are a major component of the synovial membrane and contribute to joint destruction by the secretion of pro-inflammatory cytokines and tissue-degrading enzymes [1,2]. OA is characterized by progressive destruction of articular cartilage and bone and dysregulation of synovial function [3,4]. The OA synovial membrane (SM) is prone to inflammatory processes [5], although inflammation occurs to a lower extent than in RA.

Transcription factor activator protein (AP-) 1, a homo- or heterodimer of proteins of the *JUN/FOS*-proto-oncogene families including *JUN*, *JUNB*, *JUND*, and *FOS*, appears to be strongly involved in pro-inflammatory and pro-destructive processes in patients with RA and OA [6,7,8,9]. AP-1 showed binding activity for its cognate recognition sites in the promoters of inflammation-related cytokines and matrix-degrading enzymes [10]. Therefore, AP-1 represents one of the transcription factors thought to be critically involved in the pathogenesis of arthritides [6,11]. In general the *JUN* and *FOS* family genes can be classified as “immediate-early response” genes, as they are rapidly induced by a variety of activating agents and, on the other hand, have a very short half-life (range of minutes for both mRNA and protein) [12]. Therefore, they can be regarded as markers of recent cell activation, with clearly different biological activities of individual *JUN*/*FOS* family members [7,13].

Recently, substantial progress has been made in the identification of genetic components for RA and OA [14,15]. However, additional genetic susceptibility variants need to be identified to better understand the genetics of these diseases. Among the reported genetic factors, several single nucleotide polymorphisms (SNPs) in the promoters of functionally relevant genes have demonstrated an association with the occurrence of the respective disease. For example, associations with RA were reported for SNPs in the interleukin 1β (*IL1B)* promoter [16] and the heme oxygenase I promoter [17]. Additionally, functional sequence alterations in the promoters of growth/differentiation factor 5 and ras homolog gene family member B have shown an association with OA [18,19]. To our knowledge, no candidate association studies have been published concerning the distribution of SNPs or mutations in the *JUN* and *FOS* promoters of RA and OA patients.

Therefore, the aim of this study was to analyse SNPs in the core promoters of the proto-oncogenes *JUN*, *JUNB*, *JUND*, and *FOS* in patients with rheumatic diseases, to investigate their functional relevance, and to assess potential associations between these SNPs and the occurrence of RA or knee-OA. To focus on the most relevant SNPs, a stepwise SNP selection protocol was performed: (i) Selection of polymorphic SNPs in the initial screening population, (ii) filtering for SNPs with functional effects, (iii) selection of SNPs associated with OA or RA in the discovery cohort, and (iv) replication analysis in an independent, second case control cohort.

## 2. Results

### 2.1. Initial SNP Analysis in Rheumatoid Arthritis, Knee-Osteoarthritis, and NC Samples

In this study, SNPs in the *JUN* and *FOS* promoters with a functional relevance for their expression levels in RA or OA should have been identified. Thus, the occurrence of SNPs in the *JUN* and *FOS* family gene core promoters (i.e., up to 2 kb upstream of the start codon) were initially analysed in DNA isolated from blood or SM samples of RA, knee-OA (*n* = 10 each), and NC (*n* = 5) patients using NIRCA. Only major allele variants were detected for SNPs in the promoters of *JUNB* and *JUND.* In the *JUN* promoter, 1 SNP was detected at position -1676 (C/G, major allele: C; rs4647001, position given in regard to the start codon) in 1 RA sample and another SNP at position -617/-618 (insertion CA; major allele: Without insertion; rs4647009) in 2 RA samples. In the *FOS* promoter, 2 SNPs were detected at positions -135 (A/T; major allele: A; rs2239615) and -60 (T/C, major allele: T; rs7101). Both were in perfect LD the initial screening population. All SNPs were contained in dbSNP 130 [20].

### 2.2. Functional Analyses

To address the potential functional relevance of the identified SNPs, reporter gene assays were performed following transfection of K4IM human fibroblasts, HeLa human epithelial cells, and NIH3T3 murine fibroblasts with reporter constructs. The occurrence of SNP rs4647001 in the *JUN* promoter did not influence reporter gene expression with or without stimulation with 10 ng/ml phorbol-12-myristate 13-acetate (PMA; for 8 h; data available upon request). Regarding insertion rs4647009 (i.e., the minor allele) of the *JUN* promoter, significant modulation of reporter gene expression was detected in K4IM fibroblasts 1 and 2 d following transfection either with (downregulation by approx. 85%, *p* = 0.001 for 1 d, approx. 75%, *p* = 0.05 for 2 d) or without PMA stimulation (downregulation by approx. 70%, *p* = 0.002 for 1 d, *p* = 0.05 for 2 d), as compared to the major allele (i.e., without insertion; Figure 1a,b).

Reporter gene expression was also differentially affected by alleles containing both SNPs in the *FOS* promoter (rs2239615, rs7101): Compared to the major alleles, reporter gene expression in K4IM cells was significantly upregulated in the simultaneous presence of both minor alleles. Without PMA stimulation, reporter gene expression was upregulated by approximately 70% and 105% at 1 d and 2 d (*p* = 0.05 each), respectively. In the presence of PMA, 1 d after transfection an upregulation by 100% could be detected (*p* = 0.05), whereas reporter gene expression did not significantly differ among major and minor alleles at 2 d (Figure 1c). In the latter case, however, expression levels were strongly upregulated (by approx. 235%, *p* = 0.05) in comparison to the major allele in unstimulated K4IM cells at 1 d (Figure 1d). Absolute values of the respective experiments are presented in the Appendix A.

Roughly comparable results have been obtained using HeLa and NIH3T3 cells. SNP rs4647009 caused a significant downregulation of the *JUN* promoter-dependent reporter gene expression without (HeLa: >95%, *p* = 0.05 at both time points; NIH3T3: approx. 80% at both time points, *P*=0.04 for 1 d, *p* = 0.05 for 2 d) or with PMA stimulation (HeLa: >95%, *p* = 0.05; NIH3T3: Approx. 80%, *p* = 0.04 at both time points each) (Figure 2a and Appendix A). *FOS* promoter SNPs rs2239615 and rs7101 resulted in a considerably upregulated reporter gene expression in both the absence (HeLa: >380%, *p* = 0.05 for 1 d, >200% for 2 d; NIH3T3: ≥10-fold at both time points, *p* = 0.04 for 1 d, *p* = 0.05 for 2 d) and presence of PMA (HeLa: >130%, *p* = 0.04 for 1 d, >65% for 2 d; NIH3T3: ≥10-fold at both time points, *p* = 0.04 for 1 d, *p* = 0.05 for 2 d) (Figure 2c and Appendix A). Absolute values of the respective HeLa experiments are presented in Appendix A.

### 2.3. Allelic Distribution of Polymorphisms and Statistical Analysis within the German Cohort

To analyse the allelic distribution of the identified SNPs, genotyping was performed in RA (*n* = 298), knee-OA (*n* = 277), and NC (*n* = 484) DNA samples by single base extension and matrix-assisted laser desorption/ionization—time of flight (MALDI-TOF) multiplex analysis. Functional variants in the *FOS* promoter (rs2239615, rs7101) and *JUN* promoter (rs4647009) were found to be frequent in the studied German Caucasian populations of RA cases, knee-OA cases, and NC (Table 1). Frequencies of functional variants in NC were found to be similar to those reported for Caucasian populations in dbSNP 130.

The promoter variant rs4647009 in *JUN* was not significantly associated with either RA or OA (Table 1). Promoter variants rs2239615 and rs7101 in *FOS* were found to be in perfect LD (r^2^ = 1.0; D’ = 1.0), the minor allele of rs2239615 (T) corresponding to the minor allele of rs7101 (C). Consequently, results of association analysis were similar for rs2239615 and rs7101 and are indicated hereafter as results for rs2239615/rs7101. There was no statistically significant difference in the distribution of rs2239615/rs7101 alleles between RA cases and NC (Table 1). In contrast, there was a significant correlation of the minor allele of rs2239615/rs7101 with the susceptibility for knee-OA, carriage of the homozygous minor genotype presenting a risk (Table 1). This association was reflected by the observation of rs2239615/rs7101 not being in Hardy–Weinberg equilibrium (HWE) in OA cases, but showing accordance with HWE in controls. In a recessive model, the OR for susceptibility to knee-OA was approximately doubled for carriers of the homozygous minor allele rs2239615/rs7101 (Table 1).

### 2.4. Replication of the Genetic Association within the Finnish Cohort

For independent replication of the association of rs2239615/rs7101 with knee-OA, the distribution of rs7101 was analysed in the Finnish Health 2000 study cohort. In line with the German cohort, the minor allele of rs7101 was significantly associated in an additive model with the risk for knee-OA in the Finnish cohort (OR 1.72, 95% CI 1.2–2.5, *p* = 0.006) (Table 2). The OR for carriers of at least one minor allele was 2.0 (95% CI 1.2–3.3, *p* = 0.004).

### 2.5. Meta-Analysis of the German and Finnish Studies

In order to increase the power of the analysis and to further investigate the mode of inheritance, we performed a fixed effect meta-analysis of the German and Finnish knee-OA studies (Figure 3).

Using an additive model, we found an allelic OR of 1.2 (95% CI 1.01–1.5, *p* = 0.037) with some evidence for heterogeneity (*p* = 0.06). However, under a recessive model we found an OR of the homozygous minor allele of 2.1 (95% CI 1.25–3.42, *p* = 0.004) with no evidence of heterogeneity between studies (*p* = 0.82).

## 3. Discussion

Progressive destruction of articular cartilage and bone and enduring dysregulation of normal cellular behaviour in the SM are common features of different arthritides such as RA and OA [2,3,4,5]. Many key proteins involved in their pro-inflammatory/pro-destructive properties, e.g., chemokines/cytokines and matrix-degrading enzymes, are transcriptionally regulated by AP-1 and its subunits *JUN*, *JUNB*, *JUND*, and *FOS* [6,7,8,9,21,22]. Based on the reasonable assumption that *JUN* and *FOS* family proteins influence the development/progression of rheumatic diseases [6,11,23,24], the distribution of functionally relevant SNPs in the promoters of these proto-oncogenes and their potential association with RA or knee-OA were analysed.

Various SNPs in the promoters of disease-relevant genes have shown an association with the occurrence of RA [16,17] or OA [18,19]. However, to our knowledge, our study is the first candidate gene association study covering the allelic distribution of SNPs in the core region of the *JUN* and *FOS* family promoters of RA and OA patients.

In the present study, association of two functionally relevant SNPs in the human *FOS* promoter (rs2239615/rs7101; appearing in perfect LD) with the occurrence of knee-OA was demonstrated. Both minor alleles normally occur with an allelic frequency of 26–30% (Appendix A), thus representing the SNPs with the highest minor allele frequency (MAF) in this promoter region [20]. The other six known SNPs in the first 2 kb of the human *FOS* promoter had considerably lower MAFs (0.6–7%) [20] and were not detected in the initial SNP screening in OA and RA patients. Regarding the promoters of all analysed genes, 60 SNPs were found in dbSNP, for 16 of which MAF had been reported. The power to observe SNPs within the initial screening population was 80% for SNPs with a reported MAF >3.2% (13 SNPs in dbSNP) and 99.5% for SNPs with a reported MAF >10% (11 SNPs in dbSNP). Only four of the SNPs reported in dbSNP were identified in our study. This may be related to differences in allele frequencies among Caucasian subpopulations, as described elsewhere [25].

In comparison to the major alleles, the minor alleles of SNPs rs2239615 and rs7101 modulated promoter efficiency in reporter experiments by a marked upregulation of promoter-dependent gene expression in different cell types including human and murine fibroblasts, both with and without stimulation. After 2 d, no difference could be detected between the major and the minor *FOS* promoter alleles in PMA-stimulated K4IM human fibroblasts, which may be ascribed to the high absolute expression level reached. This indicated that the occurrence of these SNPs led to an enhanced activation of the *FOS* promoter, possibly resulting in an overexpression of *FOS* proteins compared to long-term stimulated cells carrying the major allele. Consequently, there may be a general increase in AP1 complex formation [25,26] or a shift in the composition of the respective AP1 complexes [27]. To our knowledge, evidence for the functional relevance of the aforementioned SNPs in the promoters of *JUN*/*FOS* family genes was demonstrated for the first time in this study, emphasizing the potential importance of promoter SNPs in key genes for disease pathogenesis. Using DNase I hypersensitivity data from the ENCODE project [28], both SNPs appear to be located within open chromatin regions. Chromatin immunoprecipitation data from the ENCODE project further showed that transcription factors serum response factor (SRF), 300 kDa E1A-binding protein (EP300), TATA box binding protein-associated factor (TAF) 1, hairy/enhancer of split-related with YRPW motif (HEY) 1, JUND, and POU domain class 2 transcription factor 2 (POU2F2) bound within the region of the two SNPs. Accordingly, it is likely that differential allelic binding of these or other transcription factors may contribute to differential allelic expression of *FOS*. To our knowledge, cis regulation of *FOS* by rs2239615/rs7101 has not been described before. However, effects of SNPs on cis regulatory elements outside the core promoter (i.e., 800 kb distant from rs2239615/rs7101) have been described by Myers et al. (rs2591089, rs2543359, rs935340) [29]. Therefore, the present study adds new and potentially relevant information concerning the genetic regulation of *FOS*.

Allele distribution analysis in 298 RA and 277 knee-OA patients, as well as 484 NC individuals, revealed that the minor alleles of both *FOS* promoter SNPs were significantly associated with the risk for OA. Within this population, an individual’s homozygote for the minor allele occurred two times more frequently in knee-OA than in RA or NC (10.1% vs. 5.1%), resulting in a significantly higher OR (2.12) in OA patients (*p* = 0.0086). The enhanced frequency of the identified *FOS* promoter SNP alleles in knee-OA patients, in combination with the identified potential functional effects, suggests that SNPs rs2239615 and rs7101 may indeed enhance susceptibility for knee-OA. In this context, the major allele seems to mediate a protective influence possibly due to a more limited expression of *FOS*, even following stimulation. In contrast, the minor allele seems to upregulate *FOS* expression, even in a quiescent status, possibly resulting in an enhanced expression of disease-mediating genes [1,30].

Replication of the association was performed in a Finnish cohort in a total of 620 individuals. Although the number of OA individuals was smaller than that of the German study (i.e., 72 cases), definition of the two cohorts showed many similarities. Of note, the OA phenotype within the Finnish cohort was based on a clinical diagnosis similar to the definition of the OA phenotype in the original German study group. This allowed factors such as pain to be included in the phenotype, in addition to purely radiological phenotypes, which may have been lacking clinically important factors of the disease. Analysis of the Finnish cohort also revealed the minor allele of rs7101 to be significantly related to the risk of developing OA. Meta-analysis of both studies provided supporting evidence for an association, suggesting a recessive mode of inheritance. As co-variates varied between cases and controls, we performed additional analyses to address this issue. In the German group, a female-only analysis showed increased effect size of the homo minor genotype (OR = 2.55, 95% CI 1.2–5.8, *p* = 0.019). Within the smaller group of males, we found an OR = 1.6, 95% CI 0.6-4.0, *p* = 0.398). This effect size did not differ significantly from that of females (*p* = 0.77), indicating no evidence of subgroup heterogeneity. Similarly, results were highly comparable when excluding (Table 2) or including the co-variates age and gender in the association analysis of the Finnish study (minor allele OR = 1.80, 95% CI 1.2–2.7, *p* = 0.005). Thus, the detected promoter SNPs may have contributed to *FOS*-related disease progression and joint destruction in OA in an age- and gender-independent fashion [24].

However, it has to be emphasized that the relatively small number of analysed individuals represents a limitation of this study, even though OR and *P* values, as well as the successful replication, suggest a relationship between the *FOS* promoter SNPs and the occurrence of OA. Recently, genome-wide association studies (GWAS) for OA have been reported [15]. Associations with variants in *FOS* were not reported among the top-hits of these GWAS. One explanation may be that our findings could be characteristic for a yet undefined specific subgroup of knee-OA enriched in both the German and the Finnish cohort. Future studies should further address this observation.

Due to the limited power of the study, a putative association of SNPs in the *JUN* family core promoters with the RA or OA status could not be completely excluded, although variant rs4647009 in *JUN* was not associated with RA or OA in our study. Interestingly, there was differential allelic modulation of promoter activity by this SNP and there were indications for a potential NF-κB binding site in this region, as shown using the sTRAP method [31] (Appendix A). It remains to be investigated in future studies, whether or not a genetic effect also exists for the *JUN* family promoter SNPs.

## 4. Materials and Methods

### 4.1. Patients and Tissue Samples

Within the German cohort, whole blood samples were collected from patients with RA (*n* = 298) and OA (*n* = 277) in the Departments of Orthopedics and Internal Medicine III/Division of Rheumatology and Osteology (Jena University Hospital) during routine blood withdrawal. Blood samples from normal control (NC) donors (*n* = 484) were collected by the Institutes of Transfusion Medicine (University Hospitals Jena and Leipzig) during standard blood donation. The health status was controlled by hemogram, blood pressure measurements, individual medical history, and determination of virus parameters. If DNA was not immediately isolated, blood samples were stored at −20 °C.

Informed consent was obtained from all patients and donors before blood sampling. The experiments were carried out in accordance with the relevant guidelines and regulations. The study was approved by the Ethics Committee of the Jena University Hospital in accordance with the Declaration of Helsinki. RA and OA patients were classified according to the respective criteria of the American College of Rheumatology [32,33] valid within the sample assessment period. Clinical characteristics of patients/donors are presented in the Appendix A. In all patients, the disease reached a level requiring joint replacement surgery (advanced joint destruction as assessed by radiography; categorized by the same team of clinicians), thus reflecting roughly comparable disease stages in OA patients. For the analyses, exclusively OA patients with knee joint involvement were selected. All RA patients showed several affected joints, mostly including the knees.

Replication of genetic associations was done in a Finnish cohort. This sample set was a subset (Genmets) of the Finnish Health 2000 study cohort consisting of 2173 individuals, of which 2118 passed the quality control after genotyping [34]. From the cohort of 2118 genotyped individuals, 75 individuals were clinically diagnosed by a physician as having OA in one or two knee joints, did not have previous meniscus or ligament injury, and did not have sero-positive, sero-negative, or unknown RA status. In addition, 893 potential control individuals without OA, sero-positive, sero-negative, or unknown RA status were identified. In order to closer match the replication cohort to the initial discovery cohort, control and/or OA individuals who used NSAIDs, inhaled corticosteroids, had elevated levels of rheumatoid factor (RF > 30 IU/mL), or used systemic corticosteroids were excluded, resulting in 72 knee-OA and 548 control individuals in the final replication cohort (Appendix A). However, results were virtually the same when all individuals were included (Appendix A). Caucasian origin of the samples in both cohorts was validated by self-disclosure and analysis of principal components of the genetic data.

### 4.2. DNA Preparation

In the German cohort, genomic DNA was prepared from 1059 whole blood samples. Extraction of DNA was performed using QIAamp DNA blood Mini kits or the Qiagen BioRobot EZ1 Workstation (Qiagen, Hilden, Germany) according to the manufacturer’s instructions. DNA concentrations were determined using the Nanodrop ND-1000 system (PeqLab, Erlangen, Germany).

### 4.3. Conventional PCR

Conventional PCR was performed as previously described [35]. Core promoter sequences of the *JUN* (1736 bases upstream of the start codon; divided into 2 overlapping PCR fragments), *JUNB* (1831 bases; 2 fragments), *JUND* (1629 bases; 2 fragments), and *FOS* (2008 bases; 4 fragments) were amplified using sequence-specific primers. The primers contained binding sites for the restriction enzymes *Eco* RI or *Bam* HI (cloning into the vector pUC 19 for subsequent NIRCA analyses), and *Hind* III or *Not* I (cloning into the vector pUBT-luc [36] for functional analyses), respectively. The protocols for all amplified sequences are presented in the Supplement (Appendix A). Product specificity was confirmed by agarose gel electrophoresis and fluorescent cycle sequencing.

### 4.4. Cloning of DNA Fragments

DNA fragments of each analysed sequence were cloned either as wild-type control for *non-isotopic RNase cleavage assay*-based mutation analyses or for functional promoter analyses using reporter gene constructs. Therefore, the amplified DNA sequences (restricted using *Eco* RI and *Bam* HI or *Hind* III and *Not* I; New England Biolabs, Frankfurt/Main, Germany) were cloned by a standard ligation protocol into the vectors pUC19 (Invitrogen, Darmstadt, Germany) or pUBT-luc (containing the reporter gene firefly luciferase) [36].

### 4.5. Non-Isotopic RNase Cleavage Assay

For initial SNP verification in RA and OA samples, the highly sensitive non-isotopic RNase cleavage assay (NIRCA) was applied using the MutationScreener kit (Ambion, Austin, TX, USA). The assay was performed as previously described [37].

### 4.6. Functional Analyses

Endotoxin-free pUBT-luc plasmids (reporter gene: firefly luciferase) [36] containing wild-type or SNP-containing sections of the *JUN* and *FOS* promoters were transfected into 1.5 × 10^5^ K4IM human fibroblasts, HeLa human epithelial cells, or NIH-3T3 murine embryonic fibroblasts per well (cultured in DMEM + 10% FCS, 25 mM HEPES, 100 U/mL penicillin, 100 mg/mL streptomycin, 2.5 mg/mL gentamicin; 6-well plates) using Polyfect (Qiagen) or X-tremeGENE 9 (Roche, Rotkreuz, Switzerland) transfection reagents (determined in biological duplicates or triplicates). NIH3T3 cells were transfected using 1000 ng pUBT-luc per well. Based on dose response experiments in which 100, 500, or 1000 ng pUBT-luc were transfected into K4IM cells for 24 h (Appendix A), 500 ng pUBT-luc per well were applied for subsequent transfections of K4IM and HeLa cells. As an internal transfection/normalization control, the vector pRL-CMV (containing renilla luciferase; 10 ng/well) [38] was co-transfected. One and two days after transfection, cells were lysed with passive lysis buffer (Promega), either without stimulation or following 8 h stimulation with 10 ng/mL PMA. Firefly luciferase reporter gene expression was assessed via luminescence emission measurements using the dual luciferase assay system (Promega), normalized to renilla luciferase expression, and given as relative luciferase activity (RLA; i.e., firefly relative light units (RLU) divided by Renilla RLU) or in percent of the respective control. In all cases and at each time point, enzymatic activities of firefly and renilla luciferase were clearly detectable as represented in the absolute RLA values (Figure 2 and Appendix A). Data of functional analyses are presented as means ± standard error of the mean (SEM). The Mann–Whitney U test was applied to analyse differences between luciferase expression levels. Significant differences were accepted for *p* ≤ 0.05.

### 4.7. Genotyping and Statistical Analysis

Genotyping of the German cohort was performed in RA (*n* = 298), knee-OA (*n* = 277), and NC (*n* = 484) DNA samples by single base extension and MALDI-TOF multiplex analysis applying the GenoLink system (Bruker Daltonics, Bremen, Germany). Primers and cycling conditions are shown in the Appendix A. For the design of genotyping primers, the software Calcdalton was used [39]. Reaction conditions were essentially the same as those described before [40]. Based on allelic frequencies reported in dbSNP 130 [20], there was more than 80% power to detect allelic differences of 4% and 7% for *JUN* and *FOS* SNPs, respectively [41]. Genotypes in controls were in Hardy–Weinberg equilibrium. The average genotyping call rate was >99%. Frequencies of alleles and genotypes were compared between cases and controls applying standard chi-square statistics or Fisher’s exact test, as appropriate. Heterozygous genotypes of FOS −60/−135 (rs7101/rs2239615) of knee-OA samples were independently validated by sequencing using independent PCR primers. These independent PCR primers also allowed sequencing of the PCR primer binding sites of the primers used for SNP identification and genotyping, thereby excluding genotyping errors due to interfering polymorphisms at primer binding sites.

To replicate SNPs found to be significantly associated with the German discovery cohort, an independent study of Finnish origin was used. Genotyping of the Finnish sample was done using the Illumina HumanHap 610k array (Illumina, San Diego, CA, USA). For quality filtering, individuals with a genotyping success rate of less than 95%, atypical genomic heterozygosity, gender discrepancies, or relatedness with another individual were excluded from the study (*n*_excluded_ = 55). Furthermore, valid markers had to achieve genotyping success rates of at least 95%, a minor allele frequency (MAF) of at least 1% and accordance with Hardy–Weinberg equilibrium at the *p*-value cut-off 1 × 10^−6^ or more, which was observed for 555,388 SNPs. Variant rs7101 fulfilled all quality criteria mentioned, hence, no proxy was required for the replication analysis in the Finnish sample. 

ENCODE-transcription factor data are available for public use within the UCSC genome browser. They were created by the Myers Lab at the HudsonAlpha Institute for Biotechnology and by the laboratories of Michael Snyder, Mark Gerstein, and Sherman Weissman at Yale University; Peggy Farnham at UC Davis; and Kevin Struhl at Harvard. ENCODE digital DNase I hypersensitivity clusters were made available for public use through the UCSC genome browser by the UW ENCODE group [28].

*p*-Values for accordance with Hardy–Weinberg equilibrium, and association analysis (dominant, recessive, allelic) were calculated by applying chi-square statistics or Fisher’s exact test, whenever appropriate. Association analysis including the co-variates gender and age was done using PLINK [42]. For fixed-effects meta-analysis of association studies, Mantel–Haenszel methodology was applied, as implemented in package rmeta 2.16, using R 2.15.0. Within this framework, combinability of the cohorts was analysed with Woolf’s test for heterogeneity. For comparison of the OR of subgroups, a test was performed as previously described [43].

## Figures and Tables

**Figure 1 ijms-20-01382-f001:**
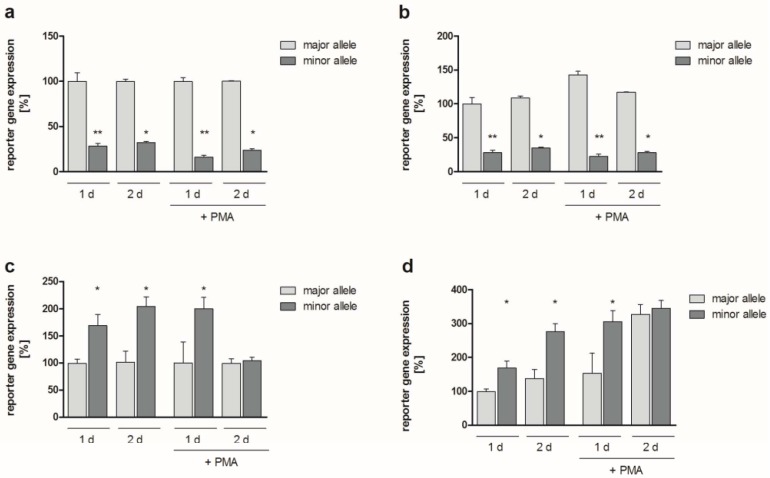
Reporter gene expression. The graph shows *JUN* (**a**,**b**) and *FOS* (**c**,**d**) promoter-dependent expression of firefly luciferase 1d and 2d following transfection of K4IM human fibroblasts, either in non-stimulated cells or in cells stimulated with 10 ng/mL PMA for 8 h (determined in biological triplicates; mean ± standard error of the mean (SEM)). Firefly luciferase expression levels were normalized to renilla luciferase expression levels in the respective samples (transfection and normalization control). Results are presented as relative values (in %) related to the expression level in the presence of the major allele at the respective time point (**a**,**c**). For better comparison, relative values related to the expression level in the presence of the major allele in unstimulated cells at 1 d are also given (**b**,**d**). ** *p* ≤ 0.005, * *p* ≤ 0.05 compared to the major allele. For *JUN*, the major allele is the rs4647009-deletion (minor allele: Insertion CA), for *FOS*, the major allele refers to the combination of rs7101-T and rs2239615-T.

**Figure 2 ijms-20-01382-f002:**
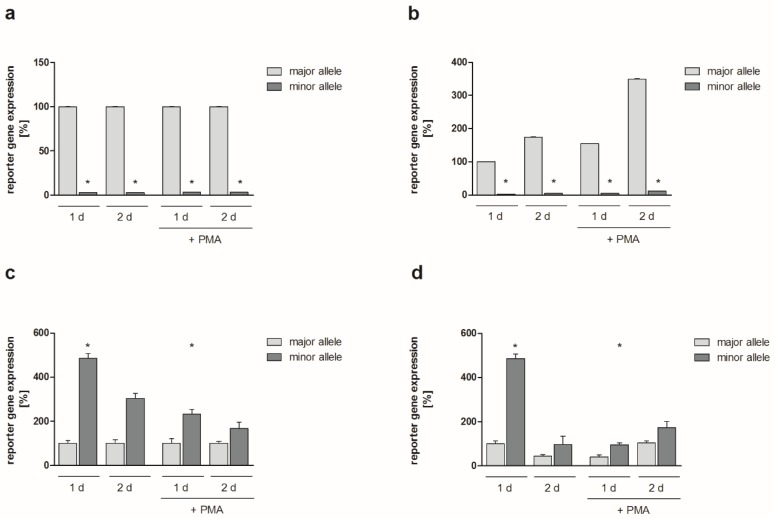
Reporter gene expression in HeLa cells. The graph shows *JUN* (**a**,**b**) and *FOS* (**c**,**d**) promoter-dependent expression of firefly luciferase 1d and 2d following transfection of HeLa human epithelial cells (±10 ng/mL PMA for 8 h, biological triplicates; in c and d, 48 h values: Biological duplicates; mean ± SEM). Firefly luciferase expression levels were normalized to renilla luciferase expression levels. Expression levels are shown as relative values (in %; **a**,**c**). For comparison, relative values related to the expression level in the presence of the major allele in unstimulated cells at 1 d are also given (**b**,**d**). * *p* ≤ 0.05 compared to the major allele.

**Figure 3 ijms-20-01382-f003:**
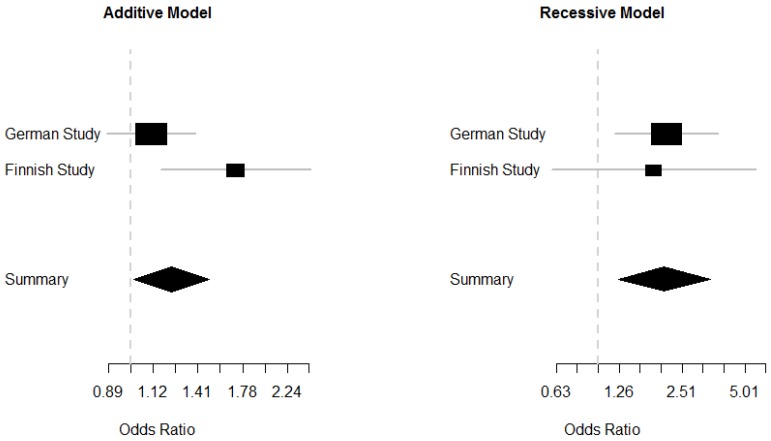
Meta-analysis of the German and Finnish study. A fixed effect meta-analysis of the German and Finnish knee-OA studies was performed under additive and recessive models of inheritance. Evidence was stronger for the recessive model (Additive model: Allelic OR = 1.2, 95% CI 1.01–1.5, *p* = 0.037, *p*_heterogeneity_ = 0.06; Recessive model: OR = 2.1, 95% CI 1.25–3.42, *p* = 0.004, *p*_heterogeneity_ = 0.82).

**Table 1 ijms-20-01382-t001:** Association analysis of rs4647009 in *JUN* and rs2239615/rs7101 in *FOS* with RA and knee-OA. Genotype counts are shown in percent (numbers). Due to the observed perfect LD between rs2239615 and rs7101, results of both SNPs are the same with allele rs7101-C corresponding to rs2239615-A, OR: Odds ratio for the minor allele, Rs4647009 is an insertion (“CA”)/deletion (“−“) polymorphism.

Variable	NC Cohort	RA Cohort	OA Cohort
**rs4647009**			
(CA/CA)	0% (0)	0% (0)	0.4% (1)
(CA/−)	11.0% (53)	11.6% (34)	9.7% (27)
(−/−)	89.0% (427)	88.4% (260)	89.9% (249)
HWE *p*-value	0.2	0.29	0.78
Allelic OR (95% CI)	/	1.05 (0.7–1.6)	0.95 (0.6–1.5)
Allelic OR *p*-value	/	0.84	0.82
**rs7101**			
Homozygous Minor (C/C)	5.0% (24)	5.1% (15)	10.1% (28)
Heterozygous (C/T)	35.1% (167)	40.3% (119)	29.2% (81)
Homozygous Major (T/T)	59.9% (285)	54.6% (161)	60.6% (168)
HWE *p*-value	0.98	0.24	0.0006
Allelic OR (95% CI)	/	1.16 (0.9–1.5)	1.13 (0.9–1.4)
Allelic OR *p*-value	/	0.23	0.35
Minor Recessive OR (95% CI)	/	1.01 (0.5–2.0)	2.12 (1.2–3.7)
Minor Recessive OR *p*-value	/	0.99	0.0086

**Table 2 ijms-20-01382-t002:** Association analysis of rs2239615/rs7101 in *FOS* with OA in the replication cohort.

Variable	NC Cohort	OA Cohort
Homozygous Minor (C/C)	17	4
Heterozygous (C/T)	184	35
Homozygous Major (T/T)	347	33
HWE *p*-value	0.26	0.23
allelic OR (95% CI)	/	1.72 (1.2–2.5)
allelic OR *p*-value	/	0.0058
Minor Recessive OR (95% CI)	/	1.84 (0.6–5.6)
Minor Recessive OR *p*-value	/	0.29

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
