# Peer review of "Association of Human *FOS* Promoter Variants with the Occurrence of Knee-Osteoarthritis in a Case Control Association Study"

_ijms, 2019, doi:10.3390/ijms20061382_

Round 1
Reviewer 1 Report
In this study, Huber et al aimed to identify SNPs in the core promotors fo JUN and FOS and to analyse whether these SNPs associate with the occurence of RA or knee OA. The data obtained from a German cohort should be validated in a Finnish cohort.
The authors identified two SNPs in JUN and another two in the FOS promotor. By using luciferase reporter assays their functional relevance was characterized. Interestingly, one SNP in JUN caused significant downregulation while both SNPs in FOS significantly upregulated reporter gene expression. Finally, they could show that the homozygous genotype of FOS promotor SNPs indeed associates with knee OA but not RA. This results could be confirmed in the Finnish cohort.
In general, I find this piece of work very interesting as it adds another susceptibility variant as a genetic component associated with the development of OA. I also strongly believe that the findings reported here are of interest for a broad readership. I also appreciate that th limitations oft he study are discussed. I only have some minor comments that could potentially further improve the quality of the mansucript:
I believe that in this case also negative results could be of interest. Therefore, I would recommend presenting all data generated in the supplement, like e.g. SNP that do not influence reproter gene expression as well as results for NIH3T3 cells.
I understand that reproter assays were pefomed in triplicates (and sometimes in duplicates). Does this mean three independet experiments or one experiment with luciferase activity being determined in three replicates. This should become also clear in the figure legends.
Why are the data for K4IM and HeLa cells presented not in the same way in panel b and d? I am uncertain if the absolute values in 2b and d can be presented as relative luciferase activity (RLA)?
Was the association analysis in the Finnish cohort also performed for JUN SNPs?
The abbreviations should be explained in the text when appearing for the first time.
Author Response
Dear Prof. Dr. Grässel and PD Dr. Aszodi,
Dear Reviewers,
Please find enclosed the revised version of our manuscript entitled “Association of Human FOS Promoter Variants with the Occurrence of Knee Osteoarthritis in a Case Control Association Study” (ijms-447789).
We thank the reviewers for the evaluation of our manuscript and their thoughtful and constructive comments. To address the concerns of the reviewers, the manuscript has been revised. Several changes and corrections have been made to the manuscript, Figures 2 and S1 have been improved, and further data (formerly noted as “data not shown”) are now illustrated in the Supplement (Figures S3, S4, and new Table S4). Together with this letter, the revised version of the manuscript has been submitted in which the changes have been highlighted in yellow. In the following, we respond point by point to the remarks of the reviewers.
Reviewer 1
Comment 1: I believe that in this case also negative results could be of interest. Therefore, I would recommend presenting all data generated in the supplement, like e.g. SNP that do not influence reporter gene expression as well as results for NIH3T3 cells.
Reply 1: We agree with the reviewer that negative results are also of interest. Unfortunately the data are stored by a colleague who does not have access to these data at present, since he is on a longer journey. Thus, a graphical representation of the negative results cannot be provided at the moment. If necessary, they could be submitted in 2-3 weeks. Alternatively, the negative results could be described in the text without graphical illustration by introducing the data with the addition “data available upon request” (page 3, line 102; highlighted in turquois).
The other results presented as “data not shown“ in the initially submitted version of our manuscript are now included as new Supplemental Figures (S2 showing reporter gene expression data in NIH3T3 cells and S3 showing the results of the sTRAP analysis) and Tables (S4 providing the association analysis of rs2239615/rs7101 in FOS with OA in the replication cohort without excluded individuals) as well as mentioned in the text (Figure S2: page 4, lines 134 and 138; Figure S3: page 8, line 280; Table S4: page 9, line 314).
Comment 2: I understand that reporter assays were performed in triplicates (and sometimes in duplicates). Does this mean three independent experiments or one experiment with luciferase activity being determined in three replicates. This should become also clear in the figure legends.
Reply 2: We apologize for the equivocal description of the reporter gene experiments. These analyses were performed in a single experimental approach for each cell line (i.e., K4IM, HeLa, and NIH3T3 cells). Within the single experiments, the experimental conditions were applied to 3 (for some conditions: 2) biological replicates. This is now clarified in the section Materials and Methods (subsection 4.6, page 9, line 348) and in the respective Figure Legends of Figure 1 (page 3, line 121-122), Figure 2 (page 4, line 143) and Figures S1 and S4 (former Figure S2; Supplement).
Comment 3: Why are the data for K4IM and HeLa cells presented not in the same way in panel b and d? I am uncertain if the absolute values in 2b and d can be presented as relative luciferase activity (RLA)?
Reply 3: As suggested by the Reviewer, panels b and d of Figure 2 have been adapted to the illustration in Figure 1, i.e., panels b and d are now presenting the relative gene expression values (in %) in relation to the expression level in the presence of the major allele at day 1 in unstimulated HeLa cells. The Figure Legend was revised accordingly (page 4, lines 145-147). Absolute reporter gene expression levels are now shown in the Supplement (revised Figure S1) to illustrate absolute differences in reporter gene expression in dependency of the different promoter sequences.
Comment 4: Was the association analysis in the Finnish cohort also performed for JUN SNPs?
Reply 4: We did not apply for a lookup of JUN-rs4647009 SNP in the Finnish sample, as our work followed the idea of a discovery-validation design to minimize multiple testing burden in the validation cohort. Hence, replication analysis was restricted to effects showing significant association in the German discovery sample. To explain our approach better in the manuscript, we rephrased the subsection describing the replication in the Materials and Methods (page 10, lines 378-387).
Comment 5: The abbreviations should be explained in the text when appearing for the first time.
Reply 5: All abbreviations are now defined in the manuscript when they are mentioned first and consistently used throughout the text. In addition, the list of abbreviations on page 11 was revised accordingly.
Reviewer 2
Comment: The authors work on an interesting field of osteoarthritis and rheumatoid arthritis research. The methodological approach, based on a case/control association study in a cohort validated in a replication cohort, is relevant. The association of 2 SNPs in FOS promoter with the occurrence of knee osteoarthritis is supported by convincing analyses.
Reply: We thank the reviewer for the positive and encouraging comment.
Please note that a few minor changes have been made throughout the manuscript to better adjust the text to the revised passages or for further clarification.
We hope that the points raised by the reviewers have been adequately addressed by the revisions as well as the point by point response, and we trust that these changes further improve the quality of our submitted work.
Yours sincerely,
René Huber
Reviewer 2 Report
The authors work on a interesting field of osteoarthritis and rheumatoid arthritis research. The methodological approach, based on a case/control association study in a cohort validated in a replication cohort, is relevant. The association of 2 SNPs in FOS promoter with the occurrence of knee osteoarthritis is supported by convincing analyses.
Author Response

(The authors gave the same response as above.)
